# TGF-β1 Activates Nasal Fibroblasts through the Induction of Endoplasmic Reticulum Stress

**DOI:** 10.3390/biom10060942

**Published:** 2020-06-22

**Authors:** Jae-Min Shin, Ju-Hyung Kang, Joo-Hoo Park, Hyun-Woo Yang, Heung-Man Lee, Ii-Ho Park

**Affiliations:** 1Upper Airway Chronic Inflammatory Diseases Laboratory, Korea University College of Medicine, Seoul 08308, Korea; shinjm0601@hanmail.net (J.-M.S.); jhkang0616@gmail.com (J.-H.K.); pjh52763@naver.com (J.-H.P.); yhw444@gmai.com (H.-W.Y.); lhman@korea.ac.kr (H.-M.L.); 2Department of Otorhinolaryngology-Head and Neck Surgery, Korea University College of Medicine, Seoul 08308, Korea

**Keywords:** transforming growth factor beta-1, unfolded protein response, extracellular matrix, airway remodeling, nose, fibroblast

## Abstract

(1) Background: Tissue remodeling and extracellular matrix (ECM) accumulation contribute to the development of chronic inflammatory diseases of the upper airway. Endoplasmic reticulum (ER) stress is considered to be the key signal for triggering tissue remodeling in pathological conditions. The present study aimed to investigate the role of ER-stress in TGF-β1-stimulated nasal fibroblasts and inferior turbinate organ cultures; (2) Methods: Fibroblasts and organ cultures were pretreated with 4-phenylbutyric acid (PBA) and stimulated with TGF-β1 or thapsigargin (TG). Expression of ER-stress markers, myofibroblast marker, and ECM components was measured by Western blotting and real-time PCR. Reactive oxygen species (ROS) were quantified using 2′,7′-dichlorofluorescein diacetate. Cell migration was evaluated using Transwell assays. Contractile activity was measured by collagen contraction assay; (3) Results: 4-PBA inhibited TGF-β1 or TG-induced ER-stress marker expression, phenotypic changes, and ECM. Pre-treatment with ROS scavengers inhibited the expression of TGF-β1-induced ER-stress markers. Migration and collagen contraction of TGF-β1 or TG-stimulated fibroblasts were ameliorated by 4-PBA treatment. These findings were confirmed in ex vivo organ cultures; (4) Conclusions: 4-PBA downregulates TGF-β1-induced ER-stress marker expression, migration, and collagen contraction via ROS in fibroblasts and organ cultures. These results suggest that ER-stress may play an important role in progression of chronic upper airway inflammatory diseases by aiding pathological tissue remodeling.

## 1. Introduction

Chronic rhinosinusitis (CRS) is defined as the inflammation of nasal and paranasal sinuses and is characterized by nasal obstruction, rhinorrhea and olfactory dysfunction [1]. The pathogenesis of CRS is not completely understood; however, several pieces of evidence indicate an important role of tissue remodeling similar to that in other chronic airway diseases such as chronic obstructive pulmonary disease and asthma [2,3]. Tissue remodeling—in response to an inflammatory insult—is a dynamic process involving matrix production and degradation. Currently, tissue remodeling is considered to be a suitable target during the treatment of CRS for two crucial reasons. First, tissue remodeling can induce irreversible structural changes that are known to be responsible for the refractoriness of CRS [4]. Second, tissue remodeling appears to occur in parallel with, rather than purely subsequent to, inflammation, thereby compromising the normal protective functions of structural cells right from the early stages of the disease [5].

Endoplasmic reticulum (ER) stress refers to a state of disrupted ER homeostasis leading to compromised protein folding. In response to ER stress, cells activate a series of signal transduction cascades, collectively termed as the unfolded protein response (UPR) [6]. Several insults—including nutrient deprivation, changes in calcium concentration, failure of post-transcriptional modifications and alterations in the oxidation-reduction balance—lead to protein misfolding in the ER [7].

Transforming growth factor (TGF)-β1 is a multifunctional growth factor with immune-modulatory and fibrogenic potential, and is known to be upregulated in chronic sinus disease right from the early stages [5]. TGF-β1 is known to increase ROS production and to inhibit the expression of antioxidant enzymes, leading to a redox imbalance [8]. In our previous study, we showed that TGF-β1 induces reactive oxygen species (ROS) generation that subsequently leads to myofibroblast differentiation and collagen production in nasal polyp-derived fibroblasts [9]. Based on this premise, we hypothesized that the activation of nasal fibroblasts in response to TGF-β1-induced ER stress occurs in a redox homeostasis-dependent manner. We investigated the ability of TGF-β1 to induce the expression of ER-stress markers such as GRP78 and XBP-1 via ROS production in nasal fibroblasts, and evaluated the potential of ER stress to induce the activation of nasal fibroblasts by assessing extracellular matrix (ECM) production and cell migration.

## 2. Materials and Methods

### 2.1. Reagents and Antibodies

Recombinant human TGF-β1 protein was obtained from R&D Systems (Minneapolis, MN, USA). 4-phenylbutyrate (4-PBA), *n*-acetyl-L-cysteine (NAC), ebselen, diphenyliodonium (DPI), thapsigargin 000000(TG), and 2′,7′-dichlorofluorescein diacetate (DCFH-DA) were procured from Sigma (St. Louis, MO, USA). Primary antibodies against GRP78, fibronectin, α-SMA, and β-actin were obtained from Santa Cruz Biotechnology (Santa Cruz, CA, USA), and XBP-1s antibody was obtained from Cell Signaling Technology (Danvers, MA, USA). HRP-conjugated anti-rabbit and anti-mouse secondary antibodies were obtained from Vector Laboratories (Burlingame, CA, USA).

### 2.2. Samples from Inferior Turbinate

Six nasal inferior turbinates were recruited from the Department of Otorhinolaryngology, Korea University Medical Center, Korea. Informed consent was obtained from each patient, and the study was approved by the Korea University Medical Center Institutional Review Board (KUGH15333-001). None of the patients had a history of allergy, asthma, or aspirin sensitivity.

### 2.3. Nasal Fibroblast Culture

Nasal fibroblasts were isolated from nasal inferior turbinate tissues by enzymatic digestion with collagenase (500 U/mL, Sigma), hyaluronidase (30 U/mL, Sigma), and DNase (10 U/mL, Sigma). Nasal fibroblasts were cultured in Dulbecco’s Modified Eagle Medium (DMEM; Invitrogen, Carlsbad, CA, USA) containing 10% fetal bovine serum (FBS; Invitrogen), 1% 10,000 units/mL penicillin, and 10,000 μg/mL streptomycin (Invitrogen) at 37 °C in 5% CO_2_. The purity of the obtained nasal fibroblasts was confirmed by the characteristic spindle-shaped morphology of the cells and flow cytometry. The fourth generation of nasal fibroblasts were used for all the experiments.

### 2.4. Organ Culture of Nasal Inferior Turbinate

The nasal inferior turbinate tissues were cut into 2 to 3 mm^3^ pieces under sterile conditions. Tissue fragments were washed three times with phosphate buffered saline (PBS). The washed tissue fragments were placed on a pre-hydrated gelatin sponge (10 mm × 10 mm × 1 mm; Spongostan, Johnson & Johnson, San Angelo, TX, USA) in 6-well plates, and filled with 1.5 mL of DMEM culture medium containing 2% FBS. The plates were maintained at 37 °C in 5% CO_2_.

### 2.5. Real-Time Polymerase Chain Reaction

Total RNA was extracted from nasal fibroblasts using TRIzol reagent (Invitrogen). cDNA was synthesized from 1 μg of RNA using MMLV reverse transcriptase (Invitrogen) according to manufacturer’s instructions. PCR was performed using the primer pairs detailed in Table 1. Gene expression analysis was performed by Quantstudio3 (Applied Biosystems, Foster City, CA, USA) using Power SYBR Green PCR Master Mix (Applied Biosystems). mRNA expression of specific genes was normalized to that of glyceraldehyde 3-phosphate dehydrogenase (*GAPDH*).

### 2.6. Western Blot

Nasal fibroblasts were lysed in PRO-PREP^TM^ (iNtRON Biotechnology, Seongnam, Korea). Protein concentrations were assessed using Bradford reagent (Bio-Rad, Hercules, CA, USA) according to manufacturer’s instructions. Equal amounts of proteins were resolved on a 10% sodium dodecyl sulfate polyacrylamide gel (SDS-PAGE), and transferred to polyvinylidene fluoride membrane (PVDF; Millipore, Billerica, MA, USA). PVDF membranes were blocked in 5% non-fat dry milk for 1 h, followed by overnight incubation with primary antibodies (GRP78, XBP-1s, α-SMA, fibronectin, β-actin) at 4 °C. Thereafter, PVDF membranes were washed with Tris-buffered saline containing Tween 20 (TBS/T buffer), and then incubated with secondary antibodies at room temperature for 1 h. Blots were visualized using the ECL system (Pierce, Rockford, IL, USA).

### 2.7. Collagen Measurement

The Sircol collagen assay (Biocolor, Belfast, UK), a quantitative dye-binding assay, which quantifies total soluble collagen in cell culture medium, was used. Briefly, nasal fibroblasts were pre-treated with 4-PBA (10 mM) for 1 h, followed by stimulation with TGF-β1 or TG for 72 h. One milliliter of sirius red, an anionic dye that reacts specifically with the basic groups in collagen side chains under assay conditions, was added to 400 mL of supernatant and incubated with gentle rotation for 30 min at room temperature. The samples were then centrifuged at 12,000 rpm for 10 min, and the collagen–dye complex precipitate was collected and re-solubilized in 0.5 M sodium hydroxide. The concentration of the dye was estimated spectrophotometrically at 540 nm (Beckman Coulter, Fullerton, CA, USA).

### 2.8. Short Interfering RNA Transfection

To analyze the mRNA and protein expression levels of XBP-1, α-SMA, fibronectin, and collagen type 1 under GRP78 knockdown conditions, nasal fibroblasts were transfected with short interfering (si)RNA directed against GRP78 (Bioneer, Daejeon, Korea) or negative control siRNA (Bioneer) according to the manufacturer’s instructions. Lipofectamine transfection reagent and GRP78 siRNA (100 nm) were mixed in Opti-MEM cell culture medium, and the cells were incubated in the mixture. Transfected cells were treated with TGF-β1 and incubated at 37 °C.

### 2.9. Measurement of Intracellular Reactive Oxygen Species

Total ROS generation was assessed using 2′,7′-DCFH-DA. Briefly, nasal fibroblasts were treated with the ROS scavengers (3 mM NAC; 10 µM ebselen; or 2 µM DPI) or 4-PBA (10 mM) for 1 h, and incubated with DCFH-DA (10 µM) in serum-free culture medium for 30 min. Thereafter, nasal fibroblasts were stimulated TGF-β1 for 2 min. The cells were washed with serum-free medium and analyzed using a fluorometer (Beckman-Coulter) and fluorescence microscope (LSM700; Zeiss, Germany).

### 2.10. Gel Contraction Assay

Rat-tail tendon collagen type I (RTTC) was purchased from BD Bioscience (Bedford, MA, USA). Nasal fibroblasts were detached using 0.05% trypsin and resuspended in serum-free DMEM. Collagen gels were prepared as previously described by mixing RTTC, serum-free DMEM and nasal fibroblasts. Nasal fibroblasts were then mixed with the neutralized collagen solution (pH 7.4) at a final cell density of 3.5 × 10^5^ cells/mL, with a final concentration of collagen at 0.75 mg/mL. Aliquots (0.5 mL/well) of the mixture of cells in collagen were cast into each well of 24-well tissue culture plates and allowed to polymerize at room temperature for around 20 min. After polymerization, the gels were gently released from the 24-well tissue culture plates and transferred into 6-well culture plate containing 1.5 mL serum free-DMEM. The gels were treated with TGF-β1 or TG with or without 4-PBA, and then incubated at 37 °C in a 5% CO_2_ atmosphere for 3 days. The area of each gel was measured using ImageJ (NIH, Bethesda, MA, USA). Data were expressed as the percentage of area compared with the initial gel area.

### 2.11. Transwell Migration Assay

Nasal fibroblasts were seeded in the upper chamber of Transwell chambers (Corning Life Sciences, MA, USA). DMEM containing 1000 U/mL penicillin and 1000 μg/mL streptomycin (Invitrogen), pre-treated with 4-PBA (10 mM) for 1 h, was added to the lower chamber of Transwell chambers. This was followed by stimulation of cells with TGF-β1 or TG for 48 h. The cells on the upper surface of the membrane were removed by cotton swabs. The cells on the lower surface of the membrane were stained with Diff-Quik stain (Sysmex, Kobe, Japan). Images of the stained cells from five selected views were captured under a microscope at 400× magnification.

### 2.12. Statistical Analysis

Significant differences between control and experimental data were analyzed using unpaired *t* test or one-way analysis of variance followed by Tukey’s test (GraphPad Prism, version 5, Graph Pad Software, San Diego, CA, USA). Significance was established at 95% confidence level. *p*-values less than 0.05 were accepted as statistically significant. Results were obtained from at least three independent experiments.

## 3. Results

### 3.1. TGF-β1 Induced the Expression of ER Stress Markers in Nasal Fibroblasts

We determined whether TGF-β1 induces ER stress in nasal fibroblasts from the inferior turbinate. We measured the levels of ER stress markers, GRP78 and XBP-1s. Nasal fibroblasts treated with TGF-β1 (5 ng/mL) were assessed for the mRNA levels of GRP78 and XBP-1s for 24 h using RT-PCR, and the corresponding protein levels for 72 h using Western blotting. We observed the ability of TGF-β1 to induce the expression of the indicated ER stress markers at the transcript and protein levels in a time-dependent manner (Figure 1).

### 3.2. 4-PBA Inhibited TGF-Β1-Induced ER Stress in Nasal Fibroblasts

To verify whether ER stress enhances the expression of GRP78 and XBP-1s, we investigated the effects of 4-PBA, a chemical chaperone that prevents ER stress in TGF-β1-induced nasal fibroblasts. Prior to experiments, an MTT assay was performed on the nasal fibroblasts to examine the effects of 4-PBA on cell survival. Serial dilutions of cells and MTT reagent were used to generate a cell titration curve. Cells were examined after treatment with 4-PBA concentrations ranging from 0 to 40 mM, and cell survival was found not to be affected by concentrations below 20 mM (data not shown). After pre-treatment with 4-PBA (2.5‒10 mM) for 1 h, cells were treated with TGF-β1 for 24 h, and the expression of GRP78 and XBP-1s mRNAs was measured. The corresponding protein levels were evaluated after 48 h. Pre-treatment with 4-PBA reduced TGF-β1-induced expression of GRP78 and XBP-1s at both mRNA and protein levels (Figure 2). We also observed that pre-treatment with 4-PBA inhibited the expression of GRP78 and XBP-1s induced by TG (2 µM), a well-known ER-stress inducer.

### 3.3. 4-PBA Inhibited TGF-Β1- or TG-Induced Phenotypic Changes in Nasal Fibroblasts

TGF-β1 induces the differentiation of nasal fibroblasts into myofibroblasts, which represent the active form of fibroblasts that synthesize ECM components [10]. We assessed whether ER stress is related to phenotypic changes in fibroblasts. We examined the expression of α-SMA—as a marker of myofibroblast differentiation—fibronectin and total soluble collagen to determine the production of ECM components.

As a first step, we evaluated the effects of 4-PBA-mediated ER stress inhibition on the phenotype of fibroblasts and ECM production in nasal fibroblasts stimulated with TGF-β1. After pre-treatment with 4-PBA (2.5‒10 mM) for 1 h, cells were treated with TGF-β1 and the expression of α-SMA, fibronectin and collagen type I or total soluble collagen was assessed. mRNA level was measured in 24 h using RT-PCR and the corresponding protein level was estimated in 72 h by Western blotting or ELISA. Pre-treatment with 4-PBA inhibited TGF-β1-induced phenotypic changes in fibroblasts as well as ECM production in nasal fibroblasts.

Next, we determined the effect of TG-induced ER stress in mediating the phenotypic change in fibroblasts. We treated cells with 2 µM TG and observed that TG induces the differentiation of nasal fibroblasts into myofibroblasts, as well as ECM production in a manner similar to TGF-β1. Pretreatment with 4-PBA also inhibited TG-induced phenotypic changes in fibroblasts and subsequent ECM production (Figure 3).

### 3.4. Regulation Of GRP78 Inhibited the Expression of ECM Components in Nasal Fibroblasts

To determine the relationship between GRP78 and ECM components, we performed the si-RNA transfection using si-GRP78. Fibroblasts were transfected with si-GRP78 (100 nM) and lipofectamine.

After 4 h, cells were stimulated with TGF-β1 (5 ng/mL) for 24–72 h. In si-control groups, TGF-β1 treatment induced the expression of GRP78, α-SMA and fibronectin. In si-GRP78 transfected groups, TGF-β1 treatment blocked the expression of α-SMA and fibronectin in nasal fibroblasts (Figure 4A,B).

### 3.5. Effect Of NAC, Ebselen and DPI on TGF-Β1-Induced Nasal Fibroblasts

In our previous study, we showed that TGF-β1 treatment enhances ROS production in nasal fibroblasts, and it is well known that ROS induces ER stress. [8,9] To evaluate the relationship between TGF-β1-induced ROS production and ER stress in nasal fibroblasts, we pre-treated cells with antioxidants including NAC (3 mM), ebselen (10 μM) and DPI (2 μM) for 1 h, followed by TGF-β1 stimulation. We determined the expression of GRP78 and XBP-1s proteins 72 h after TGF-β1 treatment by Western blot analysis. Increased expression of the concerned proteins in response to TGF-β1 was found to be inhibited by all three antioxidants (Figure 5A). Then we evaluated whether 4-PBA inhibited TGF-β1-induced ROS generation using fluorescence microscopy (Figure 5B). We observed that treatment with 4-PBA does not inhibit ROS generation in response to TGF-β1 stimulation, indicating that ROS production lies upstream of ER stress in the signaling axis.

### 3.6. 4-PBA Inhibited the Migration of TGF-Β1-Stimulated Nasal Fibroblasts

Activated myofibroblasts display not only increased production of the interstitial matrix, but also increased migratory and contractile ability. Transwell migration assays were performed to evaluate the migration of cells. Compared with the control, TGF-β1 or TG treatment (for 48 h) resulted in increased number of migrated cells. However, pre-treatment with 4-PBA for 1 h compromised the number of migrated fibroblasts (Figure 6A). Collagen gel contraction assays were also performed to measure the contractility of cells. It was observed that TGF-β1 or TG treatment significantly reduced the size of collagen gels, whereas the cells pre-treated with 4-PBA showed little or no change in collagen cell size (Figure 6B). These results show that inhibition of ER stress by 4-PBA significantly reduces fibroblast activation, thereby suppressing their contractility and migratory ability.

### 3.7. 4-PBA Inhibited TGF-Β1-Induced Activation of Fibroblasts in Nasal Inferior Turbinate Organ Cultures

To determine whether the blockade of TGF-β1-induced phenotypic changes in fibroblasts by 4-PBA could be observed in nasal tissues, we repeated our experiments with inferior turbinate organ cultures. After pre-treatment with 4-PBA (10 mM) for 1 h ex vivo, nasal inferior turbinates were treated with TGF-β1 or TG, and the expression of α-SMA, fibronectin and collagen type I or total soluble collagen was measured. The mRNA level was measured in 24 h using RT-PCR, and the corresponding protein level was assessed in 72 h by Western blotting or ELISA. Results mirrored those from experiments using nasal fibroblasts. Pre-treatment with 4-PBA inhibited both TGF-β1 or TG-induced expression of α-SMA and ECM production in nasal inferior turbinate tissues (Figure 7). These results show that 4-PBA pre-treatment can ameliorate the activation of fibroblasts induced by TGF-β1 or TG in both cells and tissues of the nose.

## 4. Discussion

In the current study, we showed that TGF-β1 enhances the expression of UPR markers (XBP-1s and GRP78), and also induces fibroblast differentiation (α-SMA) and production of ECM components such as fibronectin and collagen in nasal fibroblasts. Pre-treatment with the chemical chaperone, 4-PBA, blocks not only the expression of UPR markers, but also of α-SMA and ECM components. TG, a UPR inducer, showed similar effects with respect to phenotypic changes in nasal fibroblasts as those observed with TGF-β1. We also confirmed the role of ROS in the TGF-β1-UPR-fibroblast activation axis. ROS scavengers, such as NAC, ebselen and DPI significantly inhibited TGF-β1-induced ROS production and also the expression of UPR markers. In addition, we clearly demonstrated that 4-PBA significantly downregulates TGF-β1-induced migration and collagen gel contraction in nasal fibroblasts. Taken together, we provide evidence that ROS and consequent UPR play a key role in TGF-β1-induced differentiation of nasal fibroblasts.

Inflammation is a protective response against several endogenous and exogenous stimuli. Successful inflammation results in the restoration of homeostasis [11]. Homeostasis can be achieved through a transient adaptation without any shift in the pre-set range of inflammatory response to stress. However, the normal range of homeostatic parameters often fails to restore normal physiological functions during sustained inflammation, leading to a deviation from the homeostatic set points [12]. Although such changes protect the host from dominant inflammatory insults, they can also cause malfunction of the involved tissues via tissue remodeling. Recently, accumulating evidence have demonstrated that ongoing inflammation leads to tissue remodeling, which can contribute to recalcitrance of chronic inflammatory diseases of the upper airway, such as CRS [13]. Tissue remodeling is a process of remodeling or reconstructing an existing tissue and is generally known as a repair process that heals the wound through secretion and generation of an extracellular matrix when wounded. Active fibroblast, myofibroblast, produces extracellular matrix and participate in the reparative response. However, prolonged or excessive myofibroblast activity may result in fibrosis and organ dysfunction. In chronic rhinosinusitis, it has been found that overexpression of the extracellular matrix causes tissue remodeling of nasal tissue and is associated with pathogenesis of chronic rhinosinusitis [14,15].

Among various homeostatic mechanisms, UPR forms one of the most notable defense systems identified in recent years. UPR allows the ER to respond properly to the excessive production of unfolded proteins, which could result from several conditions such as hypoxia, nutrient deprivation, infections etc. However, pathophysiological insults and inflammatory mediators chronically induce ER stress and UPR. Studies show that UPR originating from ER stress plays a crucial role in the lungs during asthma and chronic obstructive pulmonary disease [16,17]. In particular, emerging evidence suggest that ER stress can contribute to the development and aggravation of the CRS and allergic rhinitis [18,19]. Moreover, we have previously reported that ER stress induces an epithelial-to-mesenchymal transition in A549 and primary nasal epithelial cells, which in turn contributes to tissue remodeling and disease recalcitrance in CRS [20]. Thus, we hypothesized ER stress to be the principle cause of chronic upper airway inflammatory disease and clearly demonstrated the underlying mechanism in nasal fibroblasts and nasal inferior turbinate tissues.

Excessive ROS, primarily generated by the mitochondria, has been implicated as a key signaling molecule associated with various pathological airway diseases including asthma, pulmonary fibrosis and CRS [21,22,23]. An early study has proposed that ER stress is closely associated with ROS production via increasing mitochondrial ROS levels as well as cytosolic calcium levels, thereby forming a vicious cycle [24]. Hu et al. recently suggested that histopathological changes caused by excessive ROS in the respiratory tract result in pathological remodeling [25]. We have also found evidence that tissue remodeling in the upper airway in the background of inflammatory diseases involves the TGF-β1-ROS axis [9,26].

Several studies have reported the importance of ER-stress in the airway. Bleomycin or TGF-β1 stimulated lung tissue or fibroblast caused ER-stress. This caused tissue remodeling such as myofibroblast and was involved in pulmonary fibrosis [27]. In addition, in bronchial fibroblast, ER-stress induced myofibroblast and was involved in lung fibrosis [28]. Our study focuses on the pathogenesis of CRS by using the fibroblast of the upper airway, which is different from the study conducted on the upper lower airway. First, tissue remodeling in the airways involves different diseases depending on the location. In the upper airways, it is involved in CRS with or without non-polyps, and in the lower airways it is associated with diseases such as idiopathic pulmonary fibrosis, asthma and emphysema. These differences can arise from structural differences in the upper and lower airways and intrinsic tissue vulnerabilities to external environmental substances. Fundamentally, the origin of the differing embryos in the upper and lower airways may be responsible for this difference.

4-PBA (Sodium phenylbutyrate) is known to be a histone deacetylase inhibitor and chemical chaperone. It supports folding processes by interacting with hydrophobic regions in unfolded proteins and used as an ER-stress inhibitor. In a previous study, we found that inhibition of ER-stress through 4-PBA in airway epithelial cells suppresses epithelial-to-mesenchymal transition (EMT) of airway epithelial cells. Kim et al. reported that ER-stress induces MUC5AC and MUC5B expression in human nasal airway epithelial cells and 4-PBA inhibited increased MUC5AC and MUC5B [20]. Muc5AC and MUC5B are types of mucin, and an increase in mucin in the upper airway is associated with upper airway inflammatory diseases such as allergic rhinitis and chronic rhinosinusitis. We confirmed that 4-PBA inhibits ER-stress of nasal fibroblast and relieves overexpression of ECM components and accumulation of collagen. Based on this, we selected 4-PBA as a candidate for improving CRS. In the future, we will check whether suppression of ER-stress through 4-PBA affects disease improvement in the mouse model of CRS.

In the current study, we provide direct evidence that ROS scavengers inhibit TGF-β1-induced ROS production and UPR marker expression in nasal fibroblasts. This may serve as a potential therapeutic cue for inflammatory diseases of the upper airway.

## 5. Conclusions

To our knowledge, this is the first study to describe the molecular mechanisms underlying TGF-β1-induced UPR in nasal fibroblasts via ROS. Our current study suggests that ER stress may play an important role in tissue remodeling and subsequent development of disease recalcitrance in upper airway diseases such as CRS. Furthermore, basic and clinical studies are necessary to evaluate and better understand the potential of targeting the ER-UPR pathway in the treatment of upper airway inflammatory diseases.

## Figures and Tables

**Figure 1 biomolecules-10-00942-f001:**
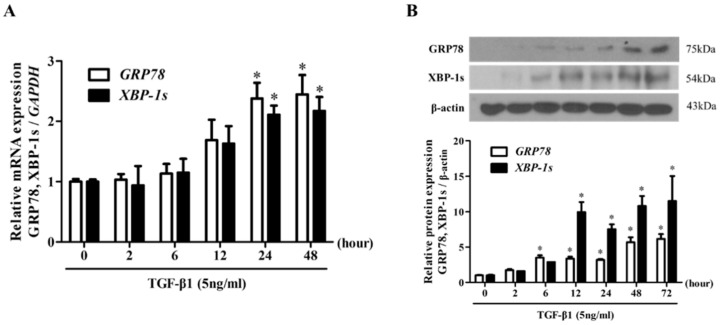
TGF-β1 upregulates the expression of ER stress-related markers in nasal fibroblasts. (**A**) Nasal fibroblasts were stimulated with TGF-β1 (5 ng/mL) for 0–48 h. The mRNA levels of ER stress-related markers, GRP78, XBP-1s, were measured by Real-time PCR. Data were normalized to glyceraldehyde 3-phosphate dehydrogenase (GAPDH) expression. (**B**) Fibroblasts were treated with TGF-β1 (5 ng/mL) for 0–72 h and the protein level of GRP78, XBP-1s was determined using Western blotting. Data were normalized to β-actin expression. Data are presented as mean ± SEM and are representative of at least three independent experiments. * *p* < 0.05 vs. control.

**Figure 2 biomolecules-10-00942-f002:**
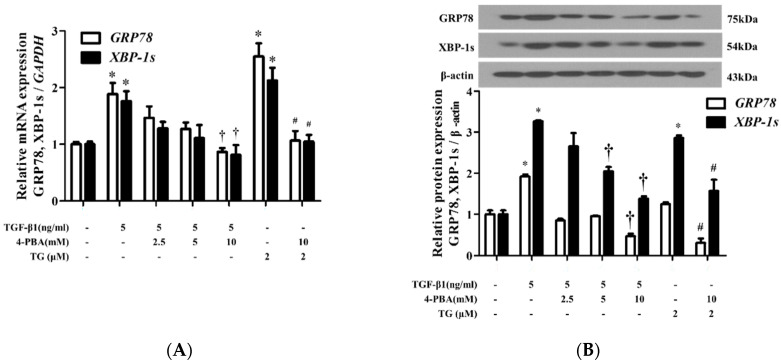
4-PBA suppresses the expression of TGF-β1-induced ER stress-related markers in nasal fibroblasts. (**A**) Nasal fibroblasts were pre-treated with 4-PBA (2.5‒10 mM) for 1 h and stimulated with TGF-β1 (5 ng/mL) for 24 h. The mRNA levels of *GRP78, XBP-1s* were measured by Real-time PCR. Data were normalized to *GAPDH* expression. (**B**) Protein levels of GRP78, XBP-1s were assessed by Western blotting after 48 h of TGF-β1 treatment. Cells treated with the ER-stress inducer, TG (2 nM) were used as a positive control. Data are presented as mean ± SEM and are representative of at least three independent experiments. * *p* < 0.05 vs. control, ^†^
*p* < 0.05 vs. TGF-β1 treatment, ^#^
*p* < 0.05 vs. TG treatment.

**Figure 3 biomolecules-10-00942-f003:**
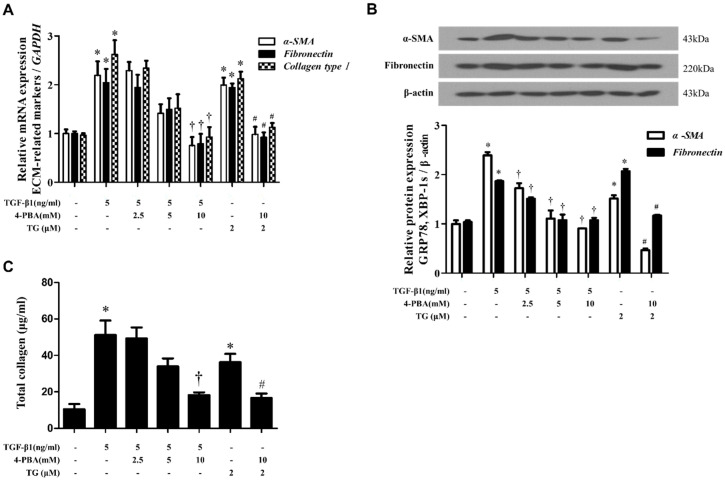
4-PBA inhibits myofibroblast differentiation and inhibits the TGF-β1-induced production of extracellular matrix components in nasal fibroblasts. (**A**) Nasal fibroblasts were pre-treated with 4-PBA (2.5‒10 mM) for 1 h and were stimulated with TGF-β1 (5 ng/mL) for 24 h. Relative mRNA expression of a-SMA, myofibroblast marker; fibronectin and collagen type I, extracellular components; was determined by Real-time PCR. Data were normalized to GAPDH levels. (**B**) Protein expression of a-SMA, fibronectin and collagen type I was analyzed by Western blotting. (**C**) Soluble collagen was determined using the Sircol assay. Cells treated with TG (2 μM) were used as a positive control. Data are presented as mean ± SEM and are representative of at least three independent experiments. * *p* < 0.05 vs. control, ^†^
*p* < 0.05 vs. TGF-β1 treatment, ^#^
*p* < 0.05 vs. TG treatment.

**Figure 4 biomolecules-10-00942-f004:**
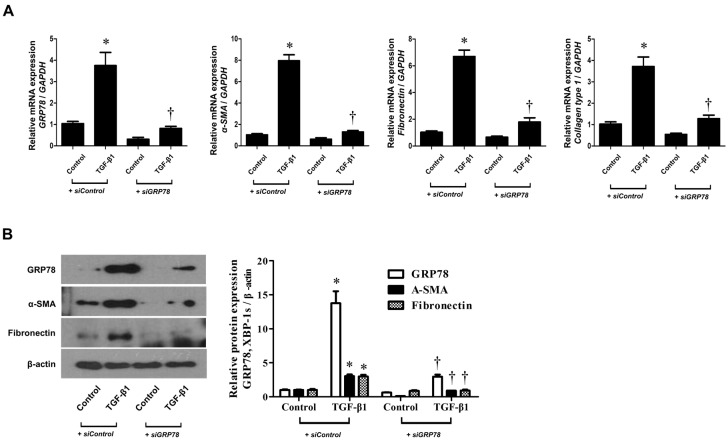
si-GRP78 suppressed the TGF- β1-induced GRP78 and ECM components in nasal fibroblasts. (**A**) Nasal fibroblasts were transfected with si-GRP78 (100 nM) for 4 h and then stimulated with TGF-β1 (5 ng/mL) for 24–72 h. The RNA levels of GRP78, α-SMA and fibronectin were determined by real-time PCR. (**B**) Protein expressions of GRP78, α-SMA and fibronectin were determined by Western blotting. Data are presented as mean ± SEM and are representative of at least three independent experiments. * *p* < 0.05 vs. si-control control, ^†^
*p* < 0.05 vs. si-control TGF-β1 treatment.

**Figure 5 biomolecules-10-00942-f005:**
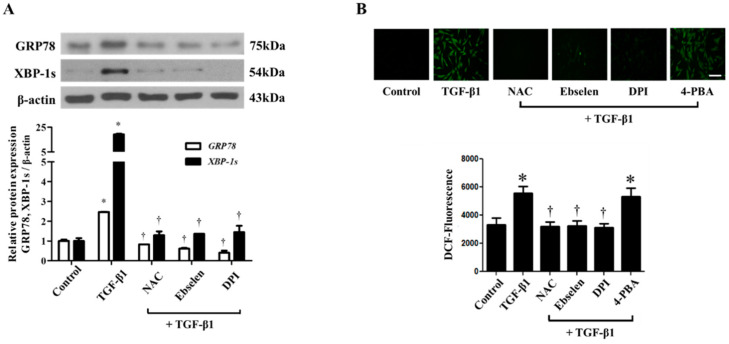
ROS scavengers, NAC, ebselen, DPI inhibit TGF- β1-induced ER stress in nasal fibroblasts. (**A**) Nasal fibroblasts were pretreated with ROS scavengers, NAC (3 mM), ebselen (10 μM), and DPI (2 μM) and then stimulated with TGF-β1 (5 ng/mL) for 72 h. The protein expression of ER stress markers, GRP78 and XBP-1s was determined by Western blotting. (**B**) ROS production was assessed by fluorescence microscopy using 2,7-dichlorofluorescein diacetate (DCFH-DA). Scale bar = 100 μm. Data are presented as mean ± SEM and are representative of at least three independent experiments. * *p* < 0.05 vs. control, ^†^
*p* < 0.05 vs. TGF-β1 treatment.

**Figure 6 biomolecules-10-00942-f006:**
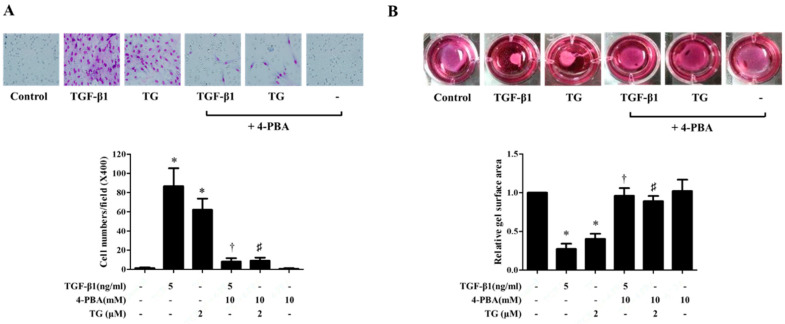
4-PBA ameliorates TGF-β1-induced migration and collagen contraction in nasal fibroblasts. Nasal fibroblasts were pre-treated with 4-PBA (10 mM) and stimulated with TGF-β1 (5 ng/mL) or TG (2 μM) for 48 h. Cells treated with TG (2 μM) were used as a positive control. (**A**) In Transwell^®^ migration assays, invasive cells were counted in five high power fields to determine the average number of migrated cells per high power field. (**B**) Collagen gel contraction was assessed by measuring the collagen gel surface area. Data are presented as mean ± SEM and are representative of at least three independent experiments. * *p* < 0.05 vs. control, ^†^
*p* < 0.05 vs. TGF-β1 treatment, ^#^
*p* < 0.05 vs. TG treatment.

**Figure 7 biomolecules-10-00942-f007:**
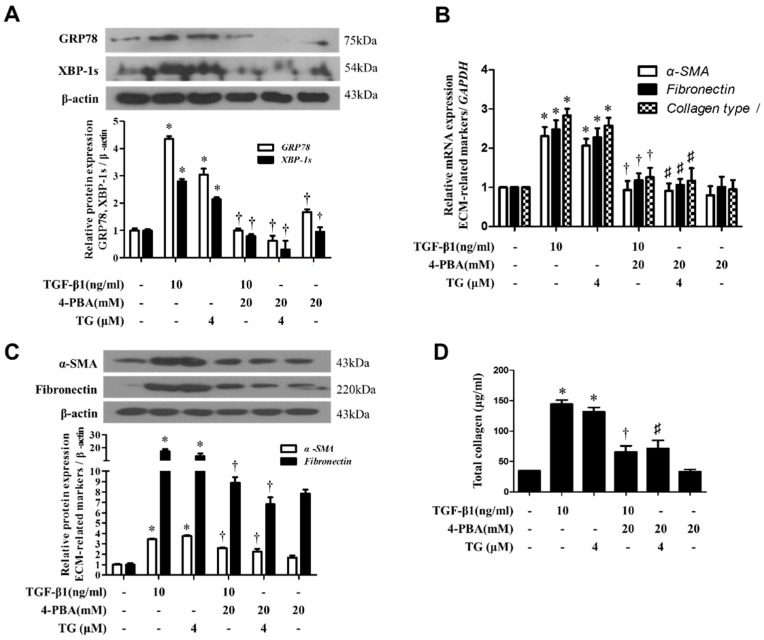
4-PBA reduces TGF-β1-induced ER-stress, myofibroblast differentiation and extracellular matrix production in ex vivo organ cultures. Inferior turbinate tissues were cultured and pre-treated with 4-PBA (20 mM) and stimulated with TGF-β1 (10 ng/mL) or TG (4 μM) for 24 or 72 h. Cells or tissues treated with TG (4 μM) were used as a positive control. (**A**-**C**) The protein and mRNA levels of ER-stress and ECM components were measured by Western blotting and real-time PCR. Data were normalized to β-actin and GAPDH levels, respectively. (**D**) Soluble collagen was determined using the Sircol assay. Data are presented as mean ± SEM and are representative of at least three independent experiments. * *p* < 0.05 vs. control, ^†^
*p* < 0.05 vs. TGF-β1 treatment, ^#^
*p* < 0.05 vs. TG treatment.

**Table 1 biomolecules-10-00942-t001:** Sequences of primers used for real-time PCR.

Primer	Direction	Sequence
*GRP78*	Forward	5′-GTT CTT GCC GTT CAA GGT GG-3′
	Reverse	5′-TGG TAC AGT AAC TGC ATG GG-3′
*XBP-1s*	Forward	5′-CCT GGT TGC TGA AGA GGA GG-3′
	Reverse	5′-CCA TGG GGA GAT GTT CTG GAG-3′
*α-smooth muscle actin*	Forward	5′-GGT GCT GTC TCT CTA GCC TCT GGA-3′
	Reverse	5′-CCC ATC AGG CAA CTC GAT ACT CTT C-3′
*Collagen type* *I*	Forward	5′-CAT CAC CTA CCA CTG CAA GAA C-3′
	Reverse	5′-ACG TCG AAG CCG AAT TCC-3′
*Fibronectin*	Forward	5′-GGA TGC TCC TGC TGT CAC-3′
	Reverse	5′-CTG TTT GAT CTG GAC CTG CAG-3′
*GAPDH*	Forward	5′-GTG GAT ATT GTT GCC ATC AAT GAC C-3′
	Reverse	5′-GCC CCA GCC TTC ATG GTG GT-3′

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
