# Peer review of "TGF-β1 Activates Nasal Fibroblasts through the Induction of Endoplasmic Reticulum Stress"

_biomolecules, 2020, doi:10.3390/biom10060942_

Round 1

Reviewer 1 Report

Overall, the work is interesting and important, but contains some flaws as it is presented, including experimental settings, and premature data interpretations with insufficient evidence. Specific comments are as follows:

  1. Fig 1B showed increasing GRP78 and XBP-1s protein expressions when treated with TGF-β1. However, it is hard to confirm the validity of data when internal control (β-actin) is not equal expression. Furthermore, Fig 1B, 2B, 3B, 4A and 6B. The densities of the bands should be quantitated with a computerized densitometer and use internal control for calibration.

  1. According to the title, the authors should confirm that the role of ER stress in TGF-β1 induced activation of fibroblasts in nasal inferior turbinate organ cultures. As shown in Fig 6, the authors did not present any significant markers for detecting ER stress, such as GRP78 and XBP-1s.

  1. Hyun Ah Baek, Do Sung Kim et al., showed the first evidence implicating the UPR (ER stress) in myofibroblastic differentiation during pulmonary fibrosis. (Am J Respir Cell Mol Biol Vol 46, Iss. , pp 731–739, Jun 2012). In the previous study, their findings and study design are very similar to present research. I strongly suggest that authors should claim what is the novelty and specific outcomes in this research and describe the relevant studies in the discussion

  1. ER stress has been reported to have relationships with multiple diseases. The authors claimed that this is the first study to describe the molecular mechanisms underlying TGF-β1-induced UPR in nasal fibroblasts via ROS. Although cells have been treated with different inhibitors to investigate the effect of ER stress in the progression of chronic rhinosinusitis, it would be more specificity by silencing the related gene of ER stress.   

  1. The authors should describe the role of 4-pba in the research, especially in the discussion. There is no description of 4-pba effect in chronic rhinosinusitis.

  1. What is tissue remodeling? In the manuscript, the authors mentioned it many times, but I still can’t clearly understand which results were used to prove tissue remodeling. The authors need to explain and give more descriptions.

Author Response

Dear reviewer,

Thank you for your feedback on our manuscript. We appreciate your constructive criticism, which has helped us to improve the manuscript. We have attempted to carefully and thoroughly address your concerns. As well, below we provide a point-by-point response to the comments.

  1. Fig 1B showed increasing GRP78 and XBP-1s protein expressions when treated with TGF-β1. However, it is hard to confirm the validity of data when internal control (β-actin) is not equal expression. Furthermore, Fig 1B, 2B, 3B, 4A and 6B. The densities of the bands should be quantitated with a computerized densitometer and use internal control for calibration.

Response: Thank you for your comment. We agree with your opinion. We conducted additional experiment. In result, we confirmed that GRP78 and XBP-1 increased significantly when internal control (β-actin) was expressed equally. Additionally, by quantifying all Western blot results, the protein expression level can be viewed numerically. All changes were reflected in the manuscript.

  1. According to the title, the authors should confirm that the role of ER stress in TGF- β1 induced activation of fibroblasts in nasal inferior turbinate organ cultures. As shown in Fig 6, the authors did not present any significant markers for detecting ER stress, such as GRP78 and XBP-1s.

Response: Thank you for your good feedback. We confirmed the expression of ER-Stress-related markers (GRP-78, XBP-1) at the tissue level using inferior turbinate organ culture. TGF-β1 and thapsigargin (TG) caused ER-Stress of nasal inferior turbinate tissues, and 4-PBA inhibited the changes induced by TGF-β1 and TG. The result attached to manuscript.

Figure 1. 4-PBA reduces TGF-β1 or TG-induced ER-Stress in ex vivo organ cultures. Inferior turbinate tissues were cultured and pre-treated with 4-PBA (20 mM) and stimulated with TGF-β1 (10 ng/ml) or TG (4 μM) for 72 hours. Cells or tissues treated with TG (4 μM) were used as a positive control. Protein levels of ER-Stress-related markers were measured by western blotting. Data were normalized to β-actin evels. Data are presented as mean ± SEM and are representative of at least three independent experiments. *p < 0.05 vs. control, †p < 0.05 vs. TGF-β1 treatment, #p < 0.05 vs. TG treatment.

  1. Hyun Ah Baek, Do Sung Kim et al., showed the first evidence implicating the UPR (ER stress) in myofibroblastic differentiation during pulmonary fibrosis. (Am J Respir Cell Mol Biol Vol 46, Iss. , pp 731–739, Jun 2012). In the previous study, their findings and study design are very similar to present research. I strongly suggest that authors should claim what is the novelty and specific outcomes in this research and describe the relevant studies in the discussion

 Response: We took a close look at the paper you mentioned. This paper is similar to our paper in that it describes that ER-Stress induced by TGF-β1 and bleomycin affects expression of a-SMA and collagen. However, the paper is an experiment confirmed in the fibroblast extracted from the lung, the lower airway. We confirmed the myofibroblast differentiation and extracellular matrix by ER-Stress in the nasal fibroblast from the upper airway. Although, the term "Unified airway" is used sometimes, the upper and lower respiratory tracts are anatomical and physiologically different organs. Role and importance of fibrosis is totally different between upper and lower respiratory tract. We also showed additional data differneitated from previous study that reviewer mentioned. By conducting additional si-RNA (GRP78) experiments, we confirmed that GRP78 regulates myofibroblast differentiation and expression of the extracellular matrix in the upper airway. All changes were reflected in the manuscript.

  1. ER stress has been reported to have relationships with multiple diseases. The authors claimed that this is the first study to describe the molecular mechanisms underlying TGF-β1-induced UPR in nasal fibroblasts via ROS. Although cells have been treated with different inhibitors to investigate the effect of ER stress in the progression of chronic rhinosinusitis, it would be more specificity by silencing the related gene of ER stress.

 Response: We appreciate your advice. As you mentioned, the best way to see the relationship between ER-Stress and chronic rhinosinusitis is to suppress ER-Stress itself. Therefore, we used si-RNA (GRP78) for more specific inhibition and confirmed the tissue remodeling related markers (alpha-smooth muscle actin, fibronectin). Inhibition of GRP78 with siRNA significantly reduced the TGF-β1-induced tissue remodeling markers. Data was uploaded as a Figure supplements.

Figure 2. Regulation of GRP78 ameliorated TGF-β1-induced ER-Stress in nasal fibroblast. Nasal fibroblast were cultured and transfected with si-GRP78 using lipofectamine 2000, and stimulated with TGF-β1 (5 ng/ml) for 24-72 hours. The mRNA and protein levels of ER-Stress-related markers were measured by real-time PCR and western blotting. Data were normalized to GAPDH and β-actin levels. Data are presented as mean ± SEM and are representative of at least three independent experiments. *p < 0.05 vs. control, †p < 0.01 vs. TGF-β1 treatment.

  1. The authors should describe the role of 4-pba in the research, especially in the discussion. There is no description of 4-pba effect in chronic rhinosinusitis.

Response: 4-PBA is a chemical chaperone that supports folding processes by interacting with hydrophobic regions in unfolded proteins. Therefore, it is used as a common ER-Stress inhibitor. In several papers that studied the relationship between ER-Stress and chronic sinusitis, 4-PBA was used as an inhibitor of ER-Stress. Kim et al, reported that ER-stress induces MUC5AC and MUC5B expression in human Nasal Airway Epithelial Cells. 4-PBA inhibited increased MUC5AC and MUC5B. Increased secretions of MUC5AC and MUC5B glycoproteins are observed from most epithelia affected by inflammatory airway disease. Therefore, reduction of MUC5AC and MUC5B by 4-PBA means improvement of disease [1]. In addition, we confirmed that ER-Stress inhibition using 4-PBA inhibited epithelial-to-mesenchymal transition of airway epithelial cells in previous study. Therefore, 4-PBA is effective in improving chronic rhinosinusitis. However, the effect was not confirmed in the nasal fibroblast. Therefore, this study was conducted and additionally mentioned in discussion part.

  1. What is tissue remodeling? In the manuscript, the authors mentioned it many times, but I still can’t clearly understand which results were used to prove tissue remodeling. The authors need to explain and give more descriptions.

Response: Tissue remodeling is a process of remodeling or reconstructing an existing tissue and is generally known as a repair process that heals the wound through secretion and generation of an extracellular matrix when wounded. Active fibroblast, myofibroblast, produces extracellular matrix and participate in the reparative response. However, prolonged or excessive myofibroblast activity may result in fibrosis and organ dysfunction. Therefore, we wrote myofibroblast differentiation and extracellular matrix production related to tissue remodeling phenomenon. In many chronic rhinosinusitis, it has been found that overexpression of the extracellular matrix causes tissue remodeling of nasal tissue and is associated with pathogenesis of chronic rhinosinusitis[2, 3]. We additionally mentioned in discussion part.

Thank you.

Sincerely yours,

Ii-Ho Park, MD, PhD

Reviewer 2 Report

This interesting article used a unique model of nasal fibroblasts from patients to study the impact of ER stress in TGF-beta induced responses. The data show that TGF-beta activated markers of ER stress that was required to regulate migration, and collagen contraction. Overall, the article is well written, the data are convincing and solid. There are different concerns that need to be addressed.   

General comments

- Most of the data to assess the role of ER stress have been generated using one single approach (4-PBA). Other experimental alternatives should be used.

- Please also provide information regarding the toxicity of 4-PBA in nasal fibroblasts since high concentrations appear to have an effect (mM range). Also comment on why only the concentration of 4-PBA (>10 mM) had an effect.

- It is not clear why among all ER stress markers, RP78 and XBP-1s were chosen since a variety of markers also exist (i.e., IRE1α, PERK and ATF6α).

- Please specify whether “n” number refers to different donors or cell preparation from a single donor.

- It is stated that cells have been isolated from n=6 patients. Please provide demographics of the patients used in the study.

Other comments:

Discussion: There should be more discussion around the originality of the present findings with regard to observations made in bronchial fibroblasts and potential impact on diseases affecting the nasal and paranasal sinuses.  

Figure 1: Because the control is set a 1 with no variance, the use of ANOVA should be used unless comparison between groups was made on the raw data (mean and SD).

Figures 2A, 3A, 5A: Same comment as above regarding statistics when comparisons are made vs control.

Figures 4 and 5: In the legend TGF was used at 5 ng/ml? Is this correct? Also, it would interesting to show that TGF does indeed produce ROS that can be inhibited with the different scavengers.

Figure 6: Why were the concentrations of TGFb and 4-PBA changed to 10microM and 20 mM, respectively?

Author Response

May 24, 2020

Manuscript number: Biomolecules-749374

Title: TGF-β1 activates nasal fibroblasts through the induction of endoplasmic reticulum stress

Dear reviewer,

Thank you for your feedback on our manuscript. We appreciate your constructive criticism, which has helped us to improve the manuscript. We have attempted to carefully and thoroughly address your concerns. As well, below we provide a point-by-point response to the comments.

 General comments

  1. Most of the data to assess the role of ER stress have been generated using one single approach (4-PBA). Other experimental alternatives should be used.

Response: We appreciate your advice. We also concluded that it is necessary to further use other inhibitors to uncover the exact mechanism. So, we used si-RNA (GRP78) for more specific inhibition and confirmed the tissue remodeling related markers (alpha-smooth muscle actin, fibronectin). The result attached to manuscript.

  1. Please also provide information regarding the toxicity of 4-PBA in nasal fibroblasts since high concentrations appear to have an effect (mM range). Also comment on why only the concentration of 4-PBA (>10 mM) had an effect.

Response: Thank you for your comment. We also recognize that the concentration of mM in cell-level experiments is high. However, in most papers that suppressed ER-stress using 4-PBA, they used the concentration of mM level. In fact, low concentrations of 4-PBA did not inhibit ER-stress [1-2]. We have additionally attached the cytotoxicity of 4-PBA in nasal fibroblast. In our MTT array (5-diphenyl tetrazolium bromide) results, LPS did not show cytotoxicity up to a concentration of 20 mM. Data was uploaded as a Figure supplements.

Figure 3. Cytotoxicity effect of LPS in nasal fibroblast. Nasal fibroblast were treated with LPS (0-40mM) for 72 hours. The cytotoxicity was determined by 5-diphenyl tetrazolium bromide (MTT) assay. Data are presented as mean ± SEM and are representative of at least three independent experiments. *p < 0.05 vs. control.

  1. It is not clear why among all ER stress markers, GRP78 and XBP-1s were chosen since a variety of markers also exist (i.e., IRE1α, PERK and ATF6α).

Response: As you mentioned, ER-Stress is largely composed of IRE1α, PERK and ATF6α pathway. We examined GRP78 and each pathway molecule to see if TGF-β1 induces ER-Stress in the initial experimental stage. As a result, treatment with TGF-β1 (5ng/ml) significantly increased the expression of GRP78 and XBP-1. Therefore, we focused on the IRE1α pathway among ER-Stress pathways. Data was uploaded as a Figure supplements.

Figure 4. TGF-β1 induces ER-stress markers; GRP78 and XBP-1 expression in nasal fibroblast. Nasal fibroblasts were stimulated with TGF-β1 (5ng/ml) for 0-72 hours. Total RNA was extracted by a TRIzol reagent and semi-quantitative PCR was performed to determine the ER-stress-related mRNA (GRP78, XBP-1, CHOP, ATF4, ATF6) expression.

  1. Please specify whether “n” number refers to different donors or cell preparation from a single donor.

Response: Thanks for the comment. We explain the meaning of the number 'N' in cell and tissue experiments. N number in cell experiment means repeat experiment in one donor. We extracted fibroblasts from donors and conducted 3 replicates. In a tissue experiment, it means the number of patients. We experimented with the tissue of five donors.

  1. It is stated that cells have been isolated from n=6 patients. Please provide demographics of the patients used in the study.

Response: Thank you for your comment. We added the information in manuscript.

Other comments:

  1. Discussion: There should be more discussion around the originality of the present findings with regard to observations made in bronchial fibroblasts and potential impact on diseases affecting the nasal and paranasal sinuses.

Response: Thanks for your comments! It may seem like a similar study in terms of ER-Stress in bronchial fibroblasts and nasal fibroblasts derived from airway affect disease. However, previous studies conducted with fibroblasts extracted from the bronchus, the lower airway. We confirmed the myofibroblast differentiation and extracellular matrix by ER-Stress in the nasal fibroblast from the upper airway. Although, the term "Unified airway" is used sometimes, the upper and lower respiratory tracts are anatomical and physiologically different organs. Role and importance of fibrosis is totally different between upper and lower respiratory tract. We also showed additional data differentiated from previous study that reviewer mentioned. By conducting additional si-RNA (GRP78) experiments, we confirmed that GRP78 regulates myofibroblast differentiation and expression of the extracellular matrix in the upper airway. All changes were reflected in the manuscript.

  1. Figure 1: Because the control is set a 1 with no variance, the use of ANOVA should be used unless comparison between groups was made on the raw data (mean and SD).

Response: Thank you for your comment. We revised it.

  1. Figures 2A, 3A, 5A: Same comment as above regarding statistics when comparisons are made vs control.

Response: Thank you for your comment. We revised it.

  1. Figures 4 and 5: In the legend TGF was used at 5 ng/ml? Is this correct? Also, it would interesting to show that TGF does indeed produce ROS that can be inhibited with the different scavengers.

Response: Yes. It is correct. TGF was used at 5 ng/ml. And we showed that TGF produce ROS that can be inhibited with the different scavengers before. (Il-Ho Park, et al. Role of ROS in TGF-β1-induced Alpha Smooth-Muscle Actin and Collagen Production in Nasal Polyp-Derived Fibroblasts. Int Arch Allergy Immunol. 2012;159(3):278-86.)

  1. Figure 6: Why were the concentrations of TGFb and 4-PBA changed to 10microM and 20 mM, respectively?

Response: The reason is that the amount of drug delivered per area is different. Considering the unit volume, when the drug is treated at the same concentration as the cell, the concentration delivered to the tissue is different, so the concentration was adjusted upward as follows. In the past, when we conducted tissue culture experiments, we conducted experiments at a concentration higher than the cell concentration.

Thank you.

Sincerely yours,

Ii-Ho Park, MD, PhD

Reference

  1. Liu, S.H., et al., Chemical chaperon 4-phenylbutyrate protects against the endoplasmic reticulum stress-mediated renal fibrosis in vivo and in vitro. Oncotarget, 2016. 7(16): p. 22116-27.
  2. Kim, D.S., et al., The regulatory mechanism of 4-phenylbutyric acid against ER stress-induced autophagy in human gingival fibroblasts. Arch Pharm Res, 2012. 35(7): p. 1269-78.

Round 2

Reviewer 1 Report

  1. According to the author’s reply, I cannot find the corresponding results in the manuscript v2 and the descriptions in figure legend are different. For example, did the author add the si-RNA (GRP78) results in Figure 2? I don’t see that in manuscript v2. I can't also find Figure supplements.
  2. If the author add the new experiments, the materials and methods need to have a corresponding experimental description.
  3. The author should check figure legend thoroughly, some contents descript “Data were normalized to GAPDH expression” but in western blot, you did not descript “Data were normalized to β-actin expression”.
  4. In Figure 6 A, the quantitative results showed the wrong mark on the y-axis.
  5. Please highlight the revised content in the manuscript.

Author Response

June 5, 2020

Manuscript number: Biomolecules-749374

Title: TGF-β1 activates nasal fibroblasts through the induction of endoplasmic reticulum stress

Dear reviewer,

Thank you for your feedback on our manuscript. There seems to have been some mistakes in the revision. We checked all the comments you mentioned and revised the manuscript. Thank you.

  1. According to the author’s reply, I cannot find the corresponding results in the manuscript v2 and the descriptions in figure legend are different. For example, did the author add the si-RNA (GRP78) results in Figure 2? I don’t see that in manuscript v2. I can't also find Figure supplements.

Response: Thank you for your comment. We reorganized the added data into figure.4. Thank you.

  1. If the author add the new experiments, the materials and methods need to have a corresponding experimental description.

Response: Thank you for your comment. We modified the manuscript, including all methods and results for the added experiment.

  1. The author should check figure legend thoroughly, some contents descript “Data were normalized to GAPDH expression” but in western blot, you did not descript “Data were normalized to β-actin expression”.

Response: Thank you for your comment. We revised it and highlight the revised content in the manuscript.

  1. In Figure 6 A, the quantitative results showed the wrong mark on the y-axis.

Response: Thank you for your comment. We revised it

  1. Please highlight the revised content in the manuscript.

Response: We highlight the revised content in the manuscript.

Thank you.

Sincerely yours,

Ii-Ho Park, MD, PhD

Reviewer 2 Report

Couples of points need attention:

Figures 2-3. TGFb1 5 ng/ml is missing in the conditions where cells are co-treated with 10 mM

Discussion: 

The revised portion added to the discussion needs attention as there is a lot of repetitions (4-PBA as a stressor for example), with sentences mal constructed or hard to understand.

It is stated "we wrote myofibroblast differentiation and extracellular matrix production related to tissue remodeling phenomenon". what do you mean?

Similarly, when it is a said that "reduction in MUC5AC and MUC5B by 4-PBA means improvement of disease". You have not done an in vivo studies to make such a statement. 

It is stated that "Therefore, 4-PBA is effective in improving chronic rhinosinusitis". This statement is misleading as it wrongly suggests that 4-PBA is clinically effective which is not the case. 

The authors stated "the upper and lower respiratory tracts are not distinguished". It is not clear what they mean?

These are just examples, I would suggest to re-write the revised section by being more concise and have better scientific discussion and description.

Author Response

June 5, 2020

Manuscript number: Biomolecules-749374

Title: TGF-β1 activates nasal fibroblasts through the induction of endoplasmic reticulum stress

Dear reviewer,

Thank you for your feedback on our manuscript. There seems to have been some mistakes in the revision. We checked all the comments you mentioned and revised the manuscript. Thank you.

  1. Figures 2-3. TGFb1 5 ng/ml is missing in the conditions where cells are co-treated with 10 Mm

Response: Thank you for your comment. We revised it

  1. The revised portion added to the discussion needs attention as there is a lot of repetitions (4-PBA as a stressor for example), with sentences mal constructed or hard to understand. It is stated "we wrote myofibroblast differentiation and extracellular matrix production related to tissue remodeling phenomenon". what do you mean? Similarly, when it is a said that "reduction in MUC5AC and MUC5B by 4-PBA means improvement of disease". You have not done an in vivo studies to make such a statement. It is stated that "Therefore, 4-PBA is effective in improving chronic rhinosinusitis". This statement is misleading as it wrongly suggests that 4-PBA is clinically effective which is not the case. The authors stated "the upper and lower respiratory tracts are not distinguished". It is not clear what they mean? These are just examples, I would suggest to re-write the revised section by being more concise and have better scientific discussion and description.

Response: Thank you for your comment. Thank you very much. We tried to solve the problem you mentioned and revised the discussion part.

Thank you.

Sincerely yours,

Ii-Ho Park, MD, PhD

Round 3

Reviewer 1 Report

  1. The resolution of Figure 4 is not good enough, especially the axis marker.
  2. The marker of "+siControl" in Fig4b Western blot is not strict. 
  3. In fig4b, the author should check the quantitative of a-SMA in the Control (+siControl) group. The expression of a-SMA seems much more than other proteins.  

Author Response

Dear reviewer,

Thank you for your feedback on our manuscript. We checked all the comments you mentioned and revised the manuscript. Thank you.

1. The resolution of Figure 4 is not good enough, especially the axis marker.

2. The marker of "+siControl" in Fig4b Western blot is not strict.

- We improved the resolution of Figure.4, and modified labels such as the x-axis and y-axis so that the reader could clearly read them.

3. In fig4b, the author should check the quantitative of a-SMA in the Control (+siControl) group. The expression of a-SMA seems much more than other proteins.

- We calculated the relative value by dividing each protein levels (GRP78, α-SMA, fibronectin) by the house keeping gene β-actin level, and then correcting the control value to 1 to obtain the relative quantitative value for each group. Therefore, even if the expression of the control band of the western blot seems different, the value is constant at 1. The above method was used to see changes in protein expression for each group compared to control. However, we found some errors in the calculation process and corrected some value.